# Blood Parasites in Sympatric Vultures: Role of Nesting Habits and Effects on Body Condition

**DOI:** 10.3390/ijerph18052431

**Published:** 2021-03-02

**Authors:** Nayden Chakarov, Guillermo Blanco

**Affiliations:** 1Department of Animal Behaviour, Bielefeld University, Konsequenz 45, 33615 Bielefeld, Germany; 2Department of Evolutionary, Ecology, National Museum of Natural Sciences, CSIC. José Gutiérrez Abascal 2, 28006 Madrid, Spain; g.blanco@csic.es

**Keywords:** avian malaria, vultures, *Leucocytozoon*, *Haemoproteus*, *Plasmodium*, scavengers, transmission, anthropogenic effects, immunity, growth time, nesting sites

## Abstract

Avian haemosporidians are a common and widespread group of vector-borne parasites capable of infecting most bird species around the world. They can negatively affect host condition and fitness. Vultures are assumed to have a very low prevalence of these blood parasites, likely due to their strong immunity; however, factors contributing to variation in host exposure and susceptibility to haemosporidians are complex, and supporting evidence is still very limited. We analyzed blood samples collected from nestlings of three vulture species in Spain over 18 years, and used updated nested-PCR protocols capable of detecting all haesmosporidian cytochrome b lineages typical for diurnal birds of prey (Accipitriformes). Similarly to previous studies, we found low haemosporidian prevalence in cliff-breeding species, with *Leucocytozoon* as the only represented blood parasite genus: 3.1% in griffon vultures (*Gyps fulvus*) (*n* = 128) and 5.3% in Egyptian vultures (*Neophron percnopterus*) (*n* = 114). In contrast, the tree-breeding cinereous vulture (*Aegypius monachus*) had a substantially higher prevalence: 10.3% (*n* = 146). By far the most common lineage in Spanish scavenging raptors was the *Leucocytozoon* lineage CIAE02. No effects of nestling age and sex, or temporal trends in prevalence were found, but an effect of nest habitat (tree-nest vs. cliff-nest) was found in the griffon vulture. These patterns may be explained by a preference of vectors to forage in and around trees rather than on cliffs and wide open spaces. We found an apparent detrimental effect of haemosporidians on body mass of nestling cinereous vultures. Further research is needed to evaluate the pathogenicity of each haemosporidian lineage and their interaction with the immune system of nestlings, especially if compromised due to pollution with pharmaceuticals and infection by bacterial and mycotic pathogens.

## 1. Introduction

Avian scavengers play a key functional role in ecosystems, particularly in landscapes shaped by extensive animal husbandry [1,2,3]. Scavengers may be subject to strong anthropogenic impacts, but the estimation of their magnitude can be hindered by insufficient baseline data. This may be the case with information about the blood parasites of vultures. So far, haemosporidian parasites of vultures have been described only through microscopic analyses, which may underestimate their diversity and prevalence. Haemosporidian parasites of the genera *Plasmodium*, *Haemoproteus*, and *Leucocytozoon* commonly infect birds of all major phylogenetic branches worldwide [4]. Most bird species can be infected by several morphologically described species of blood parasites, which correspond to an even greater diversity of tentatively isolated genetic lineages, as described by barcoding-like techniques [5]. PCR-based methods for the detection and discrimination of avian blood parasites have shown to be complementary to microscopic methods, thus increasing the sensitivity for low-intensity infections and differentiating better between taxa. All haemosporidian parasites are transmitted by blood-sucking vectors, which crucially determine their distribution and local prevalence [6]. Each of the major blood parasite genera is transmitted by a separate family of dipteran insects, which have specific breeding and foraging habitat preferences: *Plasmodium* by *Culicidae* mosquitoes, *Haemoproteus* mostly by *Ceratopogonidae* biting midges and *Hippoboscidae* louse flies, and *Leucocytozoon* by *Simulidae* blackflies. Corresponding to the preferred habitats, varies also the probability of encountering infected and naïve avian hosts of specific taxa. The same applies to species assemblages, and the probability of interspecific transmission rather than between conspecifics [7,8]. Data about lineage-sharing between host species can thus be informative, not only about the range of competent hosts for a parasite lineage, but also about habitat-related interactions between the hosts and the vectors.

A scavenger diet may expose vultures to multiple pathogens in carcasses, although several mechanisms have evolved to cope with them [9,10,11]. Vultures have many necrotizing and potentially pathogenic bacteria in their food, and gastrointestinal and skin microbiome, which may be kept at bay by a potentially highly evolved and upregulated immunity [11,12]. However, this balance may be broken by pollutants such as food-born pharmaceuticals, used in livestock for fighting against pathogens and their effects [13,14,15,16]. While the occurrence and health effects of bacterial and mycotic pathogens in vultures have attracted some attention [6], the effects on immunity of these increasingly common pollutants may be also reflected by temporal trends in blood parasite prevalence.

In this study, our aim was to describe the prevalence and associated diversity and temporal trends of haemosporidian lineages in populations of three species of sympatric old world vultures: cinereous vulture (*Aegypius monachus*), griffon vulture (*Gyps fulvus*), and Egyptian vulture (*Neophron percnopterus*). In these species, the haemosporidian fauna has only been studied through microscopy, thus providing some basis for comparison [17,18]. These studies may be complemented by more recently developed molecular methods for the detection and characterization of blood parasites, potentially providing greater taxonomic resolution and sensitivity [5]. While the diet of the study species is partly shared [19,20], there are substantial differences in their ecology that might partially explain their parasite fauna [18,21]. Egyptian vultures are long-distance migrants that overwinter in sub-Saharan Africa, while the Spanish populations of cinereous and griffon vultures are resident, with regular and occasional migrations by juvenile griffon and cinereous vultures, respectively [19]. Griffon and Egyptian vultures nest respectively in colonies and exclusive territories on cliffs, while cinereous vultures nest in sparse aggregations of trees [22]. These social and nesting habits, as well as the general environmental conditions of the breeding areas, have been highlighted as influencing factors on the occurrence of blood parasites in vultures and other species [17,18,23,24]. Therefore, spatial and temporal variations in prevalence due to inter-annually changing climatic conditions and nest habitats, and their effects on vectors can be expected [25,26]. The nesting habitat can influence the body condition of hosts through the direct impact of harsh environmental conditions in montane vs. lowland areas, and due to expenditure for food resource exploitation, and their effects on nestling condition, thereby indirectly affecting their parasite loads [27,28].

We specifically tested for temporal trends in prevalence by screening blood samples of nestling vultures collected over the last two decades. The influence of nest habitat on infection by haemosporidians was assessed in the cinereous vulture by comparing nestlings from nests located in Mediterranean dry forest vs. humid montane pinewoods. We also analyzed the effects of the nest habitat of griffon vultures, which regularly breed on cliffs but occasionally occupy tree-nests built by cinereous vultures in the study area (cliffs vs. trees). Nest habitat can affect the probability of infection, which either in combination with other factors (e.g., climatic conditions in montane vs. lowland colonies), or independently, can affect nestling body condition. Therefore, we assessed whether the body condition of nestlings was associated with nest habitat and parasite infection.

## 2. Materials and Methods

### 2.1. Ethical Statement

Our study followed the ethical guidelines proposed by the Spanish Royal Decree 1205/2005 on the protection of animals used in experiments and scientific research. The study was carried out in accordance with permits from the Spanish Bird Ringing Centre (permit number: 530115 to GB) and the regional governments of Castilla y Léon (expedient, EP/CyL/282/2001-2018) and Madrid (samples collected in collaboration with authorized personnel of the Comunidad de Madrid regional government, or provided by them). All sampling procedures were approved by these entities as part of obtaining the field permits.

### 2.2. Study Areas and Species

The study was conducted in central Spain. Cinereous vultures nest in loose colonies located in the native pine forests of the Spanish Central Range. The nests sampled were located in dry pinewoods, mostly composed of maritime pine (*Pinus pinaster*), and at an elevation of about 900–1440 m a.s.l., while those built in humid montane pinewoods of Scots pine (*Pinus sylvestris*) were located at higher altitude (1300–2200 m a.s.l.; see details of the study areas and pinewoods in [29,30]). In addition, a small group of nestlings was sampled in the Sierra Norte Natural Park (southern Spain), which is composed of Mediterranean woodland and scrublands within large extensions of dehesas, and where cinereous vultures mostly nest on cork oaks (*Quercus suber*) [20]. Sampled griffon and Egyptian vultures nested on large cliffs in the major gorge systems (Natural Parks of Hoces del Río Duratón and Hoces del Río Riaza, Central Spain) and their close surroundings [31,32], as well as in small cliffs and large rocky blocks dispersed throughout the study area. Strikingly, a small population of breeding griffon vultures nest each year as isolated pairs on structures built, but often abandoned, by cinereous vultures on Maritime pines.

Since the decline in the last decades of the extensive freely grazing ruminant herds, the three vulture species increasingly depend on carcasses from intensive livestock operations, mostly factory farms of swine and poultry [20,33]. Mostly provided at supplementary feeding stations for vulture consumption, these carcasses have been identified as the source of multiple pharmaceuticals [13,14,15], deemed an underlying cause behind the oral disease documented in nestlings of the three study species [14,16,34].

### 2.3. Fieldwork

Sampling was conducted during regular field monitoring between 2001 and 2018. Nests were accessed by climbing when nestlings were feathered but without risk of fledging prematurely (50–80 days in the cases of griffon and cinereous vultures, and 45–65 days in the case of Egyptian vulture) from May to July depending on species. Nestlings were ringed, measured with a ruler (±1 mm) for wing length (mean values ± SD: 367 ± 117, 437 ± 115, 336 ± 60 mm for cinereous, griffon, and Egyptian vultures, respectively), used as proxy of age [35,36], and weighed (±1 gr) with a balance, following standard protocols [14,35]. Rank in the hatching order in double broods of Egyptian vultures was established by wing length and plumage development. At least 1 mL of blood was taken from the brachial vein and stored in 100% ethanol for molecular sexing [36,37], parasite determination (see below), and other analyses. After manipulation, vultures were released in a good state into their nests.

### 2.4. Molecular Analyses

The samples were processed at the Department of Animal Behavior of Bielefeld University, Germany. DNA was extracted following a standard phenol-chloroform protocol. Intactness, purity, and concentration of the extracted DNA was estimated by plotting on 1% agarose gel and measuring with a NanoDrop spectrophotometer (Thermo Fisher Scientific, Waltham, MA, USA). All samples were screened with a nested PCR protocol [38], developed to amplify a cytochrome b fragment of the mitochondrial genome from the widest possible range of haemosporidian parasites, including *Leucocytozoon* of raptors. *Leucocytozoon* lineages are the most prevalent haemoparasites in raptors, but these can be systematically under-detected, especially by earlier, more popular nested PCR protocols [5,39]. The PCR protocol included two amplification rounds: the first with the primers Plas1 (5′-GAG AAT TAT GGA GTG GAT GGT G-3′) and HaemNR3 (5′-ATA GAA AGA TAA GAA ATA CCA TTC-3′), followed by a second PCR using the products of the first PCR as template and the internal primers 3760F (5′-GAG TGG ATG GTG TTT TAG AT-3′) and HaemJR4 (5′-GAA ATA CCA TTC TGG AAC AAT ATG-3′). The first PCR used a temperature profile of 10 min at 95 °C followed by 20 cycles of 30 s at 95 °C, 30 s at 48, 1 min at 72 °C, concluding with 10 min at 72 °C final elongation. The same temperature profile was used for 35 cycles in the second PCR reaction. Concentrations of PCR components were according to [38]. PCR products indicating infection status were run on 2% agarose gels. Amplicons were purified with ExoSAP (Thermo Fisher Scientific) and bi-directionally sequenced on an ABI 3730 Analyzer (Applied Biosystems, Waltham, MA, USA) with a BigDye Terminator v1.1 cycle sequencing kit (Thermo Fisher Scientific, Waltham, CA, USA) using the respective two internal primers. Raw sequences with phred score > 20 were visually inspected, manually edited, and aligned in Geneious 8.1.9 (www.geneious.com), and compared with sequences of the MalAvi database, a comprehensive list of all haemosporidian genetic lineages known to date, as of 6 August 2020 [5].

### 2.5. Statistical Analysis

Population estimates of apparent prevalence and 95% CI (Wilson approach) were calculated with Epitools Epidemiological Calculators by assuming a 0.9 test-sensitivity and a 0.99 test-specificity [40].

The patterns of infection by haemosporidians in each vulture species were assessed separately by generalized linear models (GLM) with binomial error and logit link function, where the presence (1), or lack (0), of parasites of the corresponding genus (*Leucocytozoon*) was the response variable. Nestling sex (male or female) was included as an explanatory factor, while year, year^2^, and wing length (as a proxy of age) were included as covariates. In the case of cinereous vultures, the habitat condition (dry or humid) associated to pinewoods of maritime and Scots pines, respectively, was included as an explanatory factor. In the case of griffon vultures, we also considered the nesting substrate (cliff or tree; tree-nests were built in pines by cinereous vulture) as an explanatory factor. In the case of Egyptian vulture, rank in the hatching order (0 = single nestling from broods of one nestling, 1 = first-hatched nestling from double broods, 2 = second-hatched nestling from double broods) was included nested within brood size (one or two nestlings) as an explanatory factor. The same GLMs were repeated by including year as a fixed factor (instead of covariate) to evaluate inter-annual prevalence variation irrespective of temporal trends.

Factors affecting the body mass of nestlings of each species were evaluated with generalized linear mixed-effects models (normal error, identity link function), where sex and presence of haemosporidians, nesting pine species (*P. pinaster* vs. *P. sylvestris*) in the case of cinereous vulture, nesting substrate (cliff vs. tree) in the case of griffon vulture, and hatching order nested within brood size in the case of Egyptian vulture were included as fixed factors. Wing length was included as a covariate. Year was included as a random factor. Statistical analyses and model validation were performed using SPSS software v. 25 (IBM Corp., Armonk, NY, USA) [41]. As not all data were collected for all individuals of each species, sample sizes varied slightly among analyses.

## 3. Results

We analyzed samples from 146 cinereous vultures, 128 griffon vultures, and 114 Egyptian vultures, with samples sizes varying between 1 and 38 individuals per year and host species (Table 1).

In all species, the only haemosporidian parasites found belonged to the genus *Leucocytozoon*. The most common lineage was CIAE02 (Table 1, Genbank Accession Nr. HF543631). In cinereous vultures the *Leucocytozoon* lineage AEMO02 (Genbank Accession Nr. HF543617) was also detected in two cases; these nestlings showed very poor health condition, and were subsequently admitted to a wildlife recovery center. Overall prevalence varied strongly between species, with haemoparasites found in 10.3% (95% CI: 6.3–16.3) of cinereous vultures, but only 3.1% (95% CI: 1.2–7.8) of griffon vultures, and 5.3% (95% CI: 2.4–11.0) of Egyptian vultures.

The occurrence of haemosporidians showed no linear or quadratic temporal trends (Table 1), nor dependence on age (wing length) or sex of nestlings of each vulture species (binomial GLM, all *p* > 0.11). In the case of cinereous vulture, no effect of environmental conditions derived from nesting in pinewoods of different species was found (*χ*^2^ = 0.354, *df* = 1, *p* = 0.55). Nesting substrate was found to influence the presence of parasites in nestling griffon vultures, being higher in nestlings from tree-nests than in cliff-nests (*χ*^2^ = 10.934, *df* = 1, *p* < 0.001; Figure 1). No significant effects of brood size and hatching order were found in the Egyptian vulture, both separately and with hatching order nested within brood size (all *p* > 0.12). When year was included as fixed-factor, the same conclusions were obtained, together with a lack of inter-annual variation in prevalence in the three vulture species (all *p* > 0.24).

After controlling for the effect of age (wing length), nestling body mass was lower in male than in female cinereous and Egyptian vultures, but no effect of sex was detected for griffon vultures (Table 2). The presence of haemosporidians was associated with a lower body mass only in the cinereous vulture (Table 2), affecting males and females similarly (Figure 2a). In addition, body mass was lower in nestlings, with and without parasites, from nests in montane humid pinewoods of *P. sylvestris* than from nests in dry lower-altitude pinewoods of *P. pinaster* (Table 1; Figure 2b). Year of sampling, included as a random factor, did not significantly explain body mass variation in the case of cinereous vulture (*F*_8,98_ = 1.975, *p* = 0.057), but did so in the cases of griffon and Egyptian vultures (*F*_9,94_ = 4.049, *p* < 0.001 and *F*_11,93_ = 2.18835, *p* = 0.021, respectively).

In addition, we found an effect of nesting substrate on body mass of nestling griffon vultures, indicating heavier nestlings at cliff sites than from tree-nests (Table 2).

## 4. Discussion

This is the first study assessing the prevalence and diversity of haemosporidians using molecular tools in three sympatric European avian scavengers. We found that the most common, and possibly only, blood parasite genus infecting vultures in Spain is *Leucocytozoon*.

Previous studies on griffon vulture nestlings from cliff-nests found no blood parasites through microscopic examinations in comparably large samples [17,18]. This agrees with our finding of low prevalence, amounting to four *Leucocytozoon* infections among 128 nestling griffon vultures. Greiner and Mundy [42] compared the blood parasites of five African vulture species of different life stages and found that hooded vultures (*Necrocyrtes monachus)*, white-headed vultures (*Trigonoceps occipitalis*), lapped-faced vultures (*Torgos tracheliotos*), and white-backed vultures (*Gyps africanus*) all had a blood parasite prevalence of more than 30%. Lapped-faced and white-backed vultures reached more than 70% prevalence in adulthood. Meanwhile, Cape vultures (*Gyps coprotheres*) did not show any blood parasites [42]. This agrees with our results, where most cliff-nesting griffon and Egyptian vultures had very low prevalence, while tree-nesting cinereous vultures and griffon vultures had a higher prevalence already in nestlings. An obvious commonality is the breeding habitat: Cape and griffon vultures build nests in colonies on cliff faces, often at rather barren hilltops. Egyptian vultures use a similar cliff habitat, albeit in a non-colonial fashion. In contrast, cinereous, white-headed, lapped-faced, hooded, and white-backed vultures breed on trees, sometimes close to rivers and in humid montane areas [19,43].

The differences in prevalence and correspondence to breeding habitat can be most likely explained by the behavior of haemosporidian vectors. Dipteran vectors, especially simuliids transmitting *Leucocytozoon*, are capable of and occasionally do cross larger distances, from several to hundreds of kilometers, depending on the habitat and species [44]. However, vectors mostly seek hosts close to their emergence and oviposition sites, which for blackflies are flowing water bodies [44,45]. Additionally, some vector species appear to have quite specific habitat and less specific host preferences, e.g., using cavities or keeping close to shade and intact foliage, and even preferring specific heights in the trees [26,46]. At the same time, flying in hot and open areas may strongly increase the danger of desiccation without compensating with secure encounters with hosts. Additionally, the relatively early breeding phenology of griffon vultures can lead to a substantial timing mismatch between the emergence of nestlings and blackflies, and thereby prevent contact. However, nestling griffon vultures remain in the nests for a time period (about 100 days) long enough to become infected before fledging in June–August. Overall, despite the stronger olfactory and visual cues which colonies may provide, the conditions for common foraging of vectors are probably not met by most Spanish griffon and African Cape vulture colonies, generally located in high cliffs over open, arid, and windy regions. Nonetheless, when conditions are fitting, even these species can reach high prevalence of blood parasites, as has been shown for a Cape vulture colony in Potberg, South Africa, where blackfly infestation has been shown to be very high in humid years, and where all examined nestlings had *Leucocytozoon* infections [47,48]. In support of the nesting habitat hypothesis, we found that the prevalence of haemosporidians in nestling griffon vultures was lower in cliff-nests, which form the bulk in the population, than in nests built by cinereous vultures in maritime pines. Although the sample size was small, due to the low number of nests in trees each year, this result suggests a potential relationship, even when nesting in trees implies a greater spacing of the nests in loose colonies of cinereous vultures than in monospecific but dense colonies of griffon vultures on cliffs. Interestingly, the prevalence in griffon vulture nestlings raised in tree nests exceeded even the prevalence in cinereous vultures from the same habitat. In contrast, the influence of nesting habitat on infection was not evident between dry or humid pinewoods of maritime and Scots pines, respectively, where prevalence in nestling cinereous vultures was similar, although more extensive sampling might be necessary to clarify this result.

In Spain, only *Leucocytozoon* appeared to infect vulture nestlings, which is in strong contrast to Greiner and Mundy [42], who found *Haemoproteus* to be much more common than *Leucocytozoon* for vultures in southern Africa. The exact reasons for this discrepancy are hard to pinpoint. The habitat sampled in Africa potentially corresponds more closely with the demands of *Haemoproteus*-transmitting biting midges, and less so with *Leucocytozoon*-transmitting blackflies, which need flowing water to reproduce. Similarly to other top raptors and scavengers, vultures provide very good opportunities for vector-borne transmission from an early age. The long altricial life stage guarantees immobile and defenseless hosts for vectors for a long period. Vultures may even be the most extreme among raptors in this respect, with pre-fledging periods of more than 90 days and by presenting large areas of bare skin. In comparison, 20 days in an open nest in a suitable habitat are sufficient to reach *Leucocytozoon* prevalence above 40%, as is the case in some populations of common buzzard *Buteo buteo* in Europe [49]. Indeed, sitting in a solitary nest may be a surprisingly secure way to receive, directly or through vectors, symbionts and parasites from the closest infected hosts. For solitary-breeding territorial raptors the most likely already infected hosts nearby are the parents. This pathway of quasi-vertical transmission is expected to drive parasites to higher prevalence, faster evolution, and reduced virulence, and would apply to most of the non-colonial scavengers mentioned so far [50]. Social transmission in colonies is expected to have similar, albeit weakened, consequences [51]. It may be informative for the understanding of virulence evolution to examine if parasites infecting colonial *Gyps* vultures evolve differently and lead to more substantial health consequences than parasites transmitted among solitary breeding scavengers. Additionally, in many raptor species, adults are solitary and very mobile, and their dense plumage makes them less likely to acquire infections compared to nestlings. This constraint may be relaxed for many vulture species which are also gregarious in adulthood and offer more bare skin parts for the biting of vectors and repeated exchange of blood parasites. Correspondingly, Greiner and Mundy [42] found an increase in prevalence with advancing life stage in vultures. Although, we did not analyze full-grown juvenile or adult individuals, we found no effect of nestling age or sex on the probability of showing haemosporidians in any of the tested species. Additionally, our study showed that one haemosporidian lineage is by far the most common and shared between the vulture species. In previous studies, the lineage CIAE02 has been recorded mainly in Accipitriformes raptors, but appears capable of infecting members of other orders such as gulls, rails, owls, and kingfishers [14]. The network of hosts exchanging this parasite may therefore be denser than for parasites specializing on solitary territorial hosts [7].

Depending on host and environmental conditions, haemosporidians can range from relatively benign symbionts to detrimental parasites [4,52]. In general, haemosporidians can cause temporary symptom bursts and loss of body condition, which can decrease survival rates and reproductive success [4,53,54]. However, the impact of haemosporidians on the host body condition may depend on the life stage in which they were acquired (e.g., as nestling or full-grown) [4], and on the prevailing environmental conditions in the nesting habitat and year. Nutritional and environmental stressors have been highlighted to affect susceptibility to blood parasites and their virulence [55,56]. In this study, we found a negative correlation of haemosporidian infection with body mass of nestling cinereous vultures from loose breeding colonies in pinewoods, but not in colonial cliff- and tree-nesting griffon vultures or territorial Egyptian vultures exclusively nesting in cliffs. The apparent difference between the species in the impact of blood parasites may be due to the low statistical power for detecting an effect of blood parasites with so low prevalence in the latter species. The robustness and causality behind these patterns needs further investigation by using experimental manipulation or other proxies of condition for developing nestlings [35]. As used by us, body mass could be greatly affected by food recently ingested by such large species, except for the nestling griffon vultures, who invariably vomited during handling and before weighing them. The difference in body mass between infected and non-infected cinereous vultures was independent of sex, while the well-known sexual size dimorphism in weight present in most predatory raptors [57] was also evident in nestling cinereous and Egyptian vultures, but not in griffon vultures. In addition, body mass differed between nestling griffon vultures from cliff and tree nests after controlling for age (wing length). This may be potentially connected to better development in their typical cliff-nests, which are likely associated to a better microclimate and protection against environmental conditions influencing fitness [58]. More data are necessary to establish whether the difference in condition between habitats is related to haemosporidian infection status. Independent of haemosporidian infections, body mass of nestling cinereous vultures was lower in nests from montane humid pinewoods of *P. sylvestris* than in dry lower-altitude pinewoods of *P. pinaster*. This agrees with the “human pressure hypothesis,” which states that despite the harsh environmental conditions, several large raptors and vulture species often nest in remote montane habitats because they are less accessible and frequented by humans compared to lowlands [29,59]. Such behavior may have allowed individuals to avoid the human persecution of breeding adults and their nests during the last centuries [30]. Therefore, the lower body condition of nestlings at higher elevation could be a consequence of suboptimal choice of nesting site, enforced by historic human disturbance. Such a currently suboptimal ecological niche choice can have implications for the prevalence and intensity of infections, as well as for the survival and fitness of this threatened species; and its consequences for conservation deserve further research.

The study design and the low prevalence of haemosporidians do not allow us to conclude on the ability of individuals to cope with blood parasite infections. Whatever the actual influence of nest habitat, the higher prevalence makes cinereous vultures more prone to be affected by blood parasites. The lineage AEMO02 was initially described from cinereous vultures, but in the wild it has so far been recorded only in pigeons [5]. Although no previous reports of its pathogenicity exist, it is possible that the records in vultures are spillovers in untypical hosts, and lead to increased pathogenicity [60]. In fact, the only two nestlings positive for this lineage were admitted to a recovery center due to their poor body condition, and at least one of them died subsequently. Further research is required to evaluate the pathogenicity of each haemosporidian lineage in different host species, and their interaction with the immune system of nestlings when compromised due to pollution with pharmaceuticals, and infection by bacterial and mycotic pathogens [14,16,34].

## 5. Conclusions

It has previously been suggested that scavengers may have particularly potent immune systems, capable of suppressing parasite intrusion and proliferation [10,61]. Our current results do not confirm if the adaptations connected with carrion-feeding lifestyle and gut microbiome suppress transmission, or otherwise affect the blood parasite fauna. In conjunction with the patterns found in previous studies on raptor and non-raptor scavengers, it appears probable that habitat features play an important role in the transmission of blood parasites [26,62]. An additional role in the specific features of the host immunity needs to be further explored. At the same time, the development of vulture nestlings may offer a comparatively long period of mild immune reactions allowing blood parasites to persist in these populations, and potentially in interspecific transmission networks. It has been repeatedly shown that haemosporidian parasites can cause severe damage and death to non-coadapted and weakened hosts [4,52]. Therefore, in interaction with contamination, persecution, and other condition-deteriorating disturbances, the transmission of blood parasites can potentially add to the conservation risk for some scavenger species.

## Figures and Tables

**Figure 1 ijerph-18-02431-f001:**
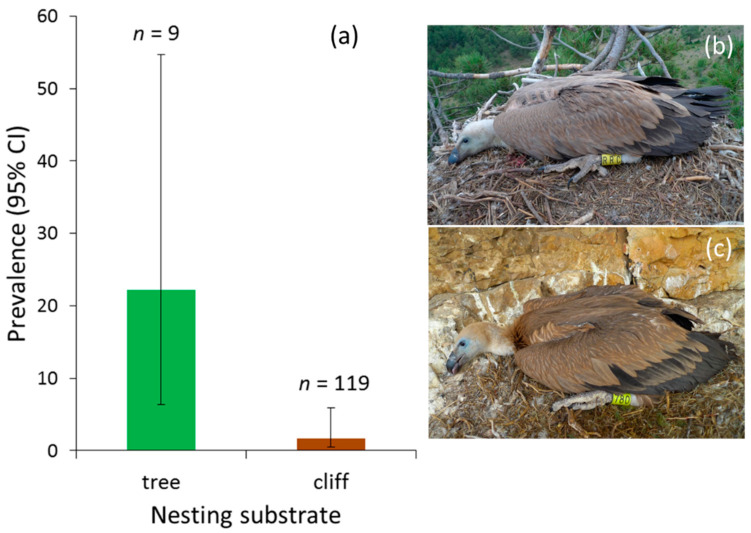
(**a**) Prevalence (±95% CI) of haemosporidians in nestling griffon vultures from nests built by cinereous vultures in maritime pines (**b**) and cliffs (**c**).

**Figure 2 ijerph-18-02431-f002:**
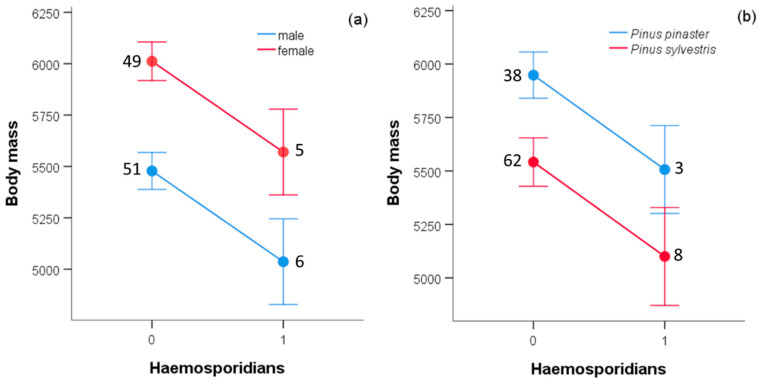
Estimated marginal means ± SE for body mass (in gr) of nestling cinereous vultures according to infection by haemosporidians, and (**a**) sex and (**b**) habitat conditions (dry or humid) associated to pinewoods and nests built in maritime pine (*P. pinaster*) and Scots Pine (*P. sylvestris*), respectively. Numbers represent sample sizes.

**Table 1 ijerph-18-02431-t001:** Total number of sampled nestlings of three vulture species with specific sample sizes given for each year (*n*), and number of infected nestlings by the corresponding *Leucocytozoon* lineage.

	*A. monachus*			*G. fulvus*		*N. percnopterus*	
Year	*n*	*CIAE02*	*AEMO02*	*n*	*CIAE02*	*n*	*CIAE02*
2001	6 *			10		–	
2004	38	4	2	–		7	
2005	10			–		11	
2006	–			–		19	1
2007	15	1		8		3	1
2008	–			–		4	1
2009	7			10	1	8	
2010	21	6		8		9	1
2011	–			5		7	
2013	5			12	1	11	1
2014	4			12		10	
2015	10	1		14	1	12	1
2016	14	1		24		–	
2017	16			24	1	13	
2018	–			1		–	
Total	146	13	2	128	4	114	6

* These nestlings marked were sampled in southern Spain, all other in central Spain.

**Table 2 ijerph-18-02431-t002:** Generalized linear models (GLMs) of body mass of nestlings of three vulture species sampled in central Spain. Parameter estimates for the levels of fixed factor and their standard errors were calculated considering the category of reference indicated in parentheses.

	Parameter Estimate				
Species, Variable	*B*	SE	*F*	df	*p*
***Aegypius monachus***					
Sex (male)	−533.619	115.171	21.467	1,98	<0.0001
Infection status (absence)	441.546	200.912	4.830	1,98	0.030
Wing length	10.895	0.688	250.508	1,98	<0.0001
Pinewood type (*P. pinaster*)	406.540	168.109	5.848	1,98	0.017
***Gyps fulvus***					
Sex (male)	−109.712	175.577	0.390	1,94	0.534
Infection status (absence)	468.791	479.693	0.955	1,94	0.331
Wing length	10.743	0.858	156.647	1,94	<0.0001
Nesting substrate (cliff)	814.339	374.061	4.739	1,94	0.032
***Neophron percnopterus***					
Sex (male)	−65.645	32.327	4.124	1,93	0.045
Infection status (absence)	60.032	70.838	0.718	1,93	0.399
Wing length	2.519	0.292	74.378	1,93	<0.0001
(Brood size = 1) (hatching order = 0)	6.866	41.758	0.015	1,93	0.985

## Data Availability

All data generated or analyzed during this study are included in this published article.

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
