# Peer review of "Blood Parasites in Sympatric Vultures: Role of Nesting Habits and Effects on Body Condition"

_ijerph, 2021, doi:10.3390/ijerph18052431_

Round 1

Reviewer 1 Report

Please see attached doc.

Author Response

See the reply attached

Reviewer 2 Report

General comments

This manuscript is a welcome contribution to our knowledge about haemosporidian parasite diversity and prevalence in of the less studied avian groups. The manuscript is rather descriptive, but the authors aimed at testing factors explaining parasite prevalence (host life history traits) and their effects on hosts’ body condition. The manuscript is well written, but in some parts of the introduction/discussion, too much emphasis is placed on some ideas and concepts that, although interesting, are not addressed here directly. Instead, other more important issues should be more elaborated. The discussion is too speculative and should focus more on the actual findings of the study, while further developing some other important aspects such as the relevance of the few lineages found and the role of vectors in the patterns found.

Specific comments

I suggest removing “temporal trends” from the title, as no trends are found.

L 10-13. The first sentence gives the impression that the work is about something different. I suggest focusing on the topic that is actually being addressed and combine the first two sentences, in something like this: “Avian haemosporidians are a common and widespread group of vector-borne parasites capable of infecting most bird species around the world and affect host condition and fitness. Vultures are assumed to have very low prevalence of these blood parasites, likely due to their strong immunity, yet factors contributing to variation in host susceptibility/exposure to haemosporidians are complex and supporting evidence is still very limited…” .

L 16. State somewhere that Leucocytozoon was the only parasite genera you found.

L 22. These patterns “may be” explained…

L 23-25. I don’t understand what “with transmission-maximizing long nestling periods of vultures” means. Do you mean that there are differences in the nesting period between the 3 species? In any case, prevalence found was generally low. Please, clarify.

L 26-28. Like the first sentence, this last statement gives a wrong impression of the study topic and overemphasizes the importance of some issues not addressed here. Please rewrite.

L 29. Revise keywords. Again, they give a wrong impression of what this study is about. It is surprising that you include Plasmodium and Haemoproteus but not Leucocytozoon (the only parasite genera found). “carcass” and “anthropogenic dependence” don’t seem very appropriate here. Please, replace with other keywords that better describe your study. In addition, “developmental time” of what? of nestlings? of parasites? in any case, doesn’t seem the right keyword either.

L 33-48. It seems you are focusing, perhaps too much, on weakened immunity. Although this is undoubtedly important, you cannot assume this is the case of all vultures. I suggest reordering the ideas to show that 1) previous studies suggest that prevalence and diversity of avian malaria and malaria-like parasites in vultures is generally low. However, evidence in support of that (or the contrary) is very limited. Only a few studies have assessed prevalence of haemosporidians in vultures and all of them are based on microscopy, which may underestimate actual prevalence values. This fact is already important in itself and your study represents an important contribution in this regard. T2) A possible explanation for this low prevalence of infection is the strong immune system of this avian group. You can develop this part and then introduce the idea that the effects of pollutants and pharmaceuticals may compromise vultures’ immunity. But this should be a marginal argument of your ms, as you are not measuring immunity nor comparing avian malaria prevalence between healthy and immunocompromised vultures, not even between individuals who eat different types of diet. It would be nice to mention other possible phylogenetic, ecological and life-history-related explanations for the low prevalence of infection in vultures.

L 56. Replace “taxons” with “taxa”

L 60. And also by Hippoboscidae louse flies

L 62. Transmit what? Host vector contact rates and their effects on parasite transmission dynamics deserve a more elaborated explanation. Use several sentences and provide appropriate references.

L 62-64. This is a key aspect in the context of your study that should be more elaborated. Please, extend these ideas and provide references.

L 67. “monachus” also in italics.

L 70-71. I believe that, roughly, foraging habits and diet of the 3 vulture species may be considered similar, and potential differences are not expected to affect exposure to vectors at all. Please, clarify what parasite fauna you refer to.

L 72-74. True, but you are sampling nestlings, so migratory behavior is not relevant in this context. I suggest focusing on the traits that may affect exposure to vectors and may in turn lead to differences in parasite prevalence between species (those that are going to be tested here). The role of insect vectors is totally omitted in this section.

L 79. What “other factors”? please, specify or rewrite.

L 81. Again, specify what factors you refer to and explain how nesting habitat or behavior (colonial vs. territorial) may influence body condition. In addition, Blanco et al. 2001 J Wildlife Dis. found no evidence for the predicted negative relationship between host body condition and intensity of parasitism. Also, clarify whether you specifically refer to relationships with parasitaemia or parasite infection.

L 85. Replace “parasitization” with “infection” by haemosporidians.

L 84. These are not differences in nest microhabitat. Mediterranean dry forest and humid montane pinewoods are largely different habitats, with different altitude, climate, flora, avifauna... Later in methods, you refer to habitat conditions (L 164). Vector populations may also greatly differ between these habitats. Explain why do you expect differences in parasite prevalence between microhabitats.

L 89. What is the expected relationship between nest microhabitat and nestling body condition? This analysis is not well justified.

L 115-120. Delete this paragraph; it is completely out of scope.

L 122. What “examined” means? Also, stating that authorized personnel handled nestlings is not necessary (it’s already said in the Ethical statement). I suggest reordering this section as: “Sampling was conducted during regular field monitoring between 2001 and 2018. Nest were accessed by climbing when nestlings were feathered but before they fledge” and indicate what is this age in days (at least the mean and range). Then “Nestlings were ringed, measured…”. Also provide more details on these methods and mean values ± SD, as they may help readers to figure out the body size/mass of sampled individuals. For example, all birds were weighed using a Pesola scale (± xx g) and the wing length was measured using a xx-cm ruler to the nearest mm. Then “Wing length was used as a proxy of age” and here provide more details on this relationship and references (if any). What is the age of the youngest nestling? This is important, as blood parasites may be undetectable in peripheral blood at very young ages.

L 127. Indicate the volume of blood sampled.

L 152. Please, indicate the quality of the reads (range). Sequencing errors and ambiguities are quite common. Did not you find any? It is also important to clarify whether DNA degradation (some samples were taken 20 years ago) may lead to sequencing errors or false-negative results. And did you detect any co-infections? Probably not given the low prevalence, but if so, how did you resolve them?

L 154. Remove full stop. 

L 160. Replace “parasitization” with “infection”. Also clarify that you run GLMs independently for each vulture species and for each parasite genus (you found a single parasite genus, but you planned to analyze them separately, right?). Add logit “link” function.

L 162-168. Indicate sample sizes of the different factors considered: sex, habitat and nesting substrate.

L 166. Remove “in the model for cinereous vulture” (already stated in the same sentence).

L 171. Did all nests of Egyptian vulture have only 1 nestling? If not, why brood size was not considered?

L 172. Add identity link “function”.

L 173. As stated in a previous comment, the relationship between nestling body mass and nesting pine species/substrate is not well justified. Why do you expect to find a relationship? Nest choice may be motivated by many different factors (age or quality of adult breeders, population density, etc.) that are far away from the questions addressed in this study. These adult- and population-related factors and others as simple as availability of food resources may lead to differences in body mass, which are likely unrelated to host-vector contact rates and hence, to parasite infection. Please, clarify this issue or remove.

L 182. Delete “only samples taken from nestlings were considered”. This was made clear before.

L 183. Explain before that you only found Leucocytozoon. These results refer exclusively to prevalence of this genus and not haemoparasites in general. Also clarify that you found only two different lineages, being CIAE02 the most prevalent one.

L 228. I wouldn’t say this study is systematic in any way. This is the first study assessing prevalence and diversity of haemosporidians using molecular tools in 3 sympatric species of avian scavengers from Spain. I would suggest authors be more modest when presenting their conclusions, taking into account that this is not a planned sampling, samples sizes are really low in some cases and some important environmental variables are not considered.

L 230. The same 3 vulture species (both adults and nestlings) were analyzed in Tella et al. 1999 PNAS, where only 2 individual A. monachus were infected by L. toddi.

 L 238. It is unclear how results from Greiner and Mundy (1979) strongly correspond to your results. Reported prevalence data were higher for all species (except Cape vultures), particularly in adult birds (you did not sample adults). In what sense do these figures correspond?

L 245. Indicate what parasite genera were found in these African vultures. Have Plasmodium ever been recorded in vultures? It would be necessary to recall that the only parasite genus you found was Leucocytozoon. Although data are scarce, you could discuss possible host-parasite associations, which is also interesting from the perspective of vectors.

L 248. Can you be more precise? How large are these distances? And in the following sentence, it is unclear what insects to refer to. Other vectors such as louse flies do not have these habitat requirements.

L 250-252. Be more precise and clearly separate the habitat requirements/foraging habits of mosquitoes and biting midges from those of blackflies.

L 255-257. What conditions are these? Could you be more precise? What about the role of variables such as the altitude of the cliffs or the phenology of different vulture species? As an example (I’m speculating) it may happen that Griffon vultures do breed before the activity of both insect vectors and parasites peaks or that the temperature in these breeding areas is still too cold… You could elaborate much more on that.   

L 264. Replace “a cause-and-effect” with “a potential relationship” between xx and xx. Your sample size is indeed too small as to suggest a causal relationship, which may be further due to by many other factors.

L 267. Replace “parasitization” with “infection”.

L 270. To clarify this potential relationship, not to fortify this result (after increasing sample size, you may found the opposite or even no relation at all).

L 272. I don't think you can affirm this, mainly considering that there are several years when no samples were obtained and that the sample sizes are extremely low in some cases, besides prevalence is very low. Were the same nests sampled in different years?

You also haven’t analyzed whether there are differences in climatic variables between those years. I would try to be more cautious in stating the existence or the lack of relationships based on the results of this study.

L 276. What other factors? Please, clarify and be more precise.

L 277. Not the most common, but the only. I would move this information, which is probably the most interesting result, above (see previous comment about haemosporidian lineages).

L 281. Is there evidence of presence/absence of louse flies in these species/regions?

L 283. Where the between-year dynamic of lineage-specific prevalence comes from? Where is this dynamic analyzed or presented?

L 285. Provide more details about the two lineages found, e.g., bird species and regions where they have been isolated (although partially unpublished, some data are shown in MalAvi). You could also comment that a third lineage L. toddi was found in A. monachus.

L 287-288. This is highly speculative. Please, delete

L 289-290. It may be that nestlings have more surface to bite on, but adults still offer quite a bit of surface susceptible to being bitten (eye rings, cere, legs...) that is still much larger than that of any other species of small birds (and we know that adults are bitten by insect vectors and often infected by blood parasites). In addition, adults may be bitten during several years, so infection probability may increase a lot during adulthood. In addition, some raptor species roost communally outside the breeding season.

L 290-292. I disagree with this logic. Please, elaborate more convincing arguments or delete.

L 295-297. It should be considered that if we add, for example, one more month of exposure in the nest, the chances of infection also increase. In addition, the phenology of vectors and parasites must be taken into account in relation to that of vultures.

L 305. OK, but it seems that prevalence in vultures is generally low, isn’t it?

L 306. Delete “maximally”

L 313-316. This is only relative. The parasite can come from many other bird species, and the vector may have bitten any other bird somewhere else before. I think all of this doesn't make much sense here. Also, you are assuming that the infection only occurs during the breeding season. Many species of raptors are migratory, including the Egyptian vulture in your study, and infection can perfectly occur during migration and/or wintering (e.g. Gutiérrez-López et al. 2015 Parasites & Vectors).

L 326-341. I would be more careful here. Your study doesn’t allow you to assume any causal relationship. An experimental approach would be needed to safely state you are showing the impact of blood parasites on host body condition. As stated above, the analysis of factors affecting body mass is not convincing (the inclusion of habitat variables in this analysis needs justification). I understand your efforts to expand the relevance of your study, but the approach taken to assess the effects of infection on host body condition is too limited and is not convincing. I suggest reconsider these analyses and interpretation of results.

L 343-352. In line with previous comment, I would remove this part. It is highly speculative and divagates around issues that are far from the topic addressed in this study.

L 353-355. The low prevalence and, most importantly, that lack of any other data evaluating the physiological condition of birds in this study.

L 360. Or may also lead to the opposite. In fact, this is again highly speculative. You cannot assume a casual relationship between the infection of this particular lineage and the poor body condition of these nestlings.

L 371. Where this connection with blood microbiome comes from? Rewrite or delete.

L 372-374. You simply don’t know.

L 375-383. This is extremely speculative. Please, delete and rewrite while trying to focus on discussing what you found.

Author Response

See the reply to the reviewer attached

Reviewer 3 Report

The manuscript by Chakarov et. al provides a survey of blood parasites in sympatric vultures in spain over 18 years. The manuscript is decently written and can be useful in the field of bird parasitology. My comments are as follows. 

  1. In terms of methods and experiments there is not a lot in the manuscript. Although, since the samples were collected over 18 years,  it makes it a valuable dataset. 
  2. Authors mention in these species, the haemosporidian fauna has only been studied through microscopy, thus making the sequencing data important.  

    The authors should deposit their sequences in a publicly available database.
  3. The authors must include a phylogenetic analysis of their sequences along with some of the previously published sequences. 

Author Response

The reply to the reviewer is attached below

Reviewer 4 Report

See the attachment.

Author Response

(The authors gave the same response as above.)

Round 2

Reviewer 2 Report

I really appreciate the effort that authors have done to address all my concerns. In general, I find the new version much clearer, making the paper far improved. Nonetheless, I still have a few comments and suggestions that I believe will help improving even more the clarity and quality of the study. The paper would also benefit from an in-depth revision of the English. For example, there are several typos throughout the ms that I have not highlighted. Revising the writing style of some sentences would help understanding your message.

Specific comments:

L 25. I suggest deleting the last part of the sentence. The pattern found is a low prevalence of infection by avian haemosporidians, and this is difficult to explain as a consequence of the long time of growth of nestling vultures. It would have sense if the prevalence of infection was very high, but not the opposite.

L 29. “…immune system of nestlings, especially if compromised due to pollution…”.

L 67. Why these effects may be reflected by “temporal” trends in parasite prevalence? Effects on immunity may also affect blood parasite prevalence, but I cannot find the link with the temporal dimension. Please, clarify.

L 76-77. Delete “while utilizing blood samples collected for host DNA analysis”.

L 88-90. This sentence is unclear. I know what you mean, but it’s not well written. For example, “…through resource availability and energy expenditure…” What is the connection between habitat and energy expenditure? And do you mean energy expenditure of adults or nestlings? This needs clarification.

L 97. What other life history traits? Please, be more specific.

L 98. …affect “nestling” body condition.

L 99. You don’t do that. You are looking for any association between -only some of the multiple variables that may be involved- and nestling body condition. As previously discussed, your study doesn’t allow establishing causal relationships. Please, rewrite into something like: “We assessed whether body condition of nestlings is associated with nest habitat and parasite infection”.

L 136. There is an extra “s”.

L 141-143. Delete this sentence as it is already stated in the first sentence of the paragraph.

L 147. Describe somewhere the protocol used for molecular sexing, at least the reference/s.

L 213. Revise format. Data shown in the Table are intermingled and is almost impossible to read.

L 265. What is exactly what strongly corresponds to your results? Differences in prevalence among species? Lower prevalence in cliff-nesting species? It is unclear. Greiner & Mundy report high prevalence of haemosporidians in all but 1 of 5 vulture species analyzed (while you report very low prevalence in all species, being higher in one of them) and these data correspond to individuals of different age classes (you analyzed only nestlings). Please, clarify.

In addition, I suggest starting a new paragraph in line 259. Explanations from line 268 onwards (new paragraph) discuss your results in relation with those found by Greiner & Mundy, and would be better connected if they appear in the same paragraph.

L 338-340. This is an apparently interesting idea but the sentence is unclear. What does “model” mean in this context? Please rewrite to clarify what you mean.

L 380. Again, your study design and the general lack of information on, for example, nestlings’ immunity, changes in prevalence within-individuals over time, etc… (and not only the low prevalence of haemosporidians) do not allow you conclude anything about the ability of individuals to cope with parasite infections. Please, rewrite or delete.

Author Response

The response to the reviewer comments was attached as a Word document.

Reviewer 3 Report

The authors have addressed the queries 

Author Response

The text have been reviewed by a native English